# Inhibition of Human Osteoclast Differentiation by Kynurenine through the Aryl-Hydrocarbon Receptor Pathway

**DOI:** 10.3390/cells10123498

**Published:** 2021-12-10

**Authors:** So-Yeon Kim, Younseo Oh, Sungsin Jo, Jong-Dae Ji, Tae-Hwan Kim

**Affiliations:** 1Institute for Rheumatology Research, Hanyang University, Seoul 04763, Korea; rlath109@naver.com (S.-Y.K.); epris12@naver.com (Y.O.); joejo0517@gmail.com (S.J.); 2Department of Translational Medicine, Graduate School of Biomedical Science and Engineering, Hanyang University, Seoul 04763, Korea; 3Department of Rheumatology, College of Medicine, Korea University, Seoul 02841, Korea; 4Department of Rheumatology, Hanyang University Hospital for Rheumatic Diseases, Seoul 04763, Korea

**Keywords:** aryl-hydrocarbon receptor, osteoclastogenesis, osteoclast, bone remodeling, kynurenine, rheumatoid arthritis

## Abstract

Aryl-hydrocarbon receptor (AhR) is a ligand-activated transcription factor and regulates differentiation and function of various immune cells such as dendritic cells, Th17, and regulatory T cells. In recent studies, it was reported that AhR is involved in bone remodeling through regulating both osteoblasts and osteoclasts. However, the roles and mechanisms of AhR activation in human osteoclasts remain unknown. Here we show that AhR is involved in human osteoclast differentiation. We found that AhR expressed highly in the early stage of osteoclastogenesis and decreased in mature osteoclasts. Kynurenine (Kyn), formylindolo[3,4-b] carbazole (FICZ), and benzopyrene (BaP), which are AhR agonists, inhibited osteoclast formation and Kyn suppressed osteoclast differentiation at an early stage. Furthermore, blockade of AhR signaling through CH223191, an AhR antagonist, and knockdown of AhR expression reversed Kyn-induced inhibition of osteoclast differentiation. Overall, our study is the first report that AhR negatively regulates human osteoclast differentiation and suggests that AhR could be good therapeutic molecule to prevent bone destruction in chronic inflammatory diseases such as rheumatoid arthritis (RA).

## 1. Introduction

Osteoclast is a multinucleated giant cell derived from hematopoietic cell of the myeloid lineage. Receptor activators of nuclear factor kappa B ligand (RANKL) and macrophage colony-stimulating factor (M-CSF) produced by osteoblasts are essential cytokines for osteoclast differentiation and functions. M-CSF promotes formation of osteoclast precursors (OCPs) that express the RANK receptor for RANKL. When binding RANKL to RANK on OCPs, it differentiates into mature osteoclasts to have potent bone resorption activity [1,2]. Excessive bone resorption causes the bone destruction characteristic of certain chronic inflammatory diseases, such as rheumatoid arthritis (RA). Thus, inhibition of osteoclasts could be beneficial for the treatment of the bone destruction in RA [3,4].

Aryl-hydrocarbon receptor (AhR) is a member of the Pern-Arnt-Sim superfamily of ligand-activated transcription factors that regulates the xenobiotic metabolism. It is well known that AhR is involved in immune and inflammatory diseases by regulating various immune cells, such as dendritic cells, Th17, and regulatory T cells [5,6,7]. In AhR intracellular signaling, the genomic and non-genomic pathway is involved. At steady state, AhR interacts with various proteins, such as the co-chaperone p23, the AhR-interacting protein (XAP2), and the heat shock protein (Hsp90) and forms a protein complex localized in the cytoplasm. Upon binding of AhR to its agonist, AhR dissociates from the protein complex, translocates into the nucleus, interacts with aryl hydrocarbon receptor nuclear translocator (ARNT), and binds to dioxin response element (DRE) sequences leading to inducing of target genes such as cytochrome P450 family 1 subfamily B member 1 (CYP1B1), aryl-hydrocarbon receptor repressor (AHRR) and TCDD inducible poly (ADP-ribose) polymerase (TIPARP). In non-genomic pathway, AhR induces phosphorylation of target proteins via c-src and works as an E3 ubiquitin ligase that targets proteins for ubiquitination and degradation by proteasome [5,6]. AhR agonists include exogenous ligands, such as 2,3,7,8-tetrachlorodibenzo-*p*-dioxin (TCDD) and benzopyrene (BaP), and endogenous ligands, such as formylindolo[3,4-b] carbazole (FICZ) and kynurenine (Kyn) [7]. Among these AhR ligands, BaP and TCDD are smoke pollutants. Smoking, known as a major factor in RA, induces bone destruction by increasing osteoclastic bone resorption [8]. Several studies have suggested that smoke contaminants, including BaP and TCDD, induce bone loss via AhR [9,10,11].

Bone remodeling occurs by bone-resorbing osteoclasts and bone-forming osteoblasts. AhR agonists have shown to inhibit osteoblast differentiation [12]. In contrast, studies of the effect of AhR on mouse osteoclast differentiation and function showed inconsistent results depending on agonist concentration, treatment time, or cell density [12,13].

Furthermore, there is no study on human osteoclasts in the context of AhR modulation. All studies about the effects of AhR on osteoclasts were conducted in mouse cells, not in human cells. Notably, there are significant differences in the regulation of osteoclast differentiation in humans and mice. [14]. Therefore, the aim of our study is to provide the exact roles of AhR on human osteoclast differentiation, compared with mice.

## 2. Materials and Methods

### 2.1. Material

Recombinant human M-CSF and soluble RANKL were purchased from PeproTech (Rocky Hill, NJ, USA). Kyn, FICZ, BaP, CH223191, MG132, and chloroquine (CQ) were purchased from Sigma-Aldrich (St. Louis, MO, USA).

### 2.2. Cell Isolation

Human peripheral blood mononuclear cells (PBMCs) were obtained from healthy donors and RA synovial fluid mononuclear cells (RA SFMCs) were obtained from synovial fluid of RA patients using Ficoll gradient centrifugation (GE Healthcare, Chicago, IL, USA). Monocytes were then isolated from the PBMCs or RA SFMCs using anti-CD14 magnetic beads (Miltenyi Biotec, Auburn, CA, USA) according to the manufacturer’s instructions. This study was approved by the Ethics Committee of Hanyang University Hospital (IRB No. 2008-09-001).

### 2.3. Osteoclast Differentiation

PBMC CD14+ cells were cultured with 20 ng/mL human M-CSF in α-MEM medium (Gibco, Gaithersburg, MD, USA) including 10% FBS (Gibco, Gaithersburg, MD, USA) and 1% penicillin/streptomycin (Gibco, Gaithersburg, MD, USA) for 2 days to generate OCPs. Additionally, the OCPs were incubated with 20 ng/mL M-CSF and 40 ng/mL RANKL for an additional 6 days. For osteoclastogenesis in RA SFMC CD14+ cells, the cells were cultured in α-MEM with 20 ng/mL M-CSF and 40 ng/mL RANKL for 9 days. The culture medium was replenished every 3 days.

Murine OCPs isolation and osteoclast differentiation were previously reported [15]. Bone marrow cells were obtained from femurs and tibia of C57BL/6 mice. Briefly, bone marrow cells were treated with RBCs lysis using ACK lysis buffer (Invitrogen, Carlsbad, CA, USA) and culture on culture dishes with 10 ng/mL murine M-CSF (Peprotech, Rocky Hill, NJ, USA) for a day. Then, suspension cells were collected and further cultured with 20 ng/mL murine M-CSF for a day, followed by treatment with 50 ng/mL M-CSF and 100 ng/mL RANKL for 4 days. The culture medium was replenished at 3 days. This study was approved by the institutional animal care and use committee of Hanyang University (HY-IACUC-2021-0170).

### 2.4. TRAP Stain and F-Actin Ring

At the end of the culture period, osteoclasts were fixed with 10% formalin and stained for acid phosphatase 5, tartrate resistant (TRAP) using kit (CosmoBio, Carlsbad, CA, USA) following the manufacturer’s recommendations. TRAP+ osteoclasts (more than three nuclei) were counted. For detection of F-actin ring formation, cells were fixed and permeabilized with 0.1% Triton X-100, and then incubated with FITC phalloidin (Sigma-Aldrich, St. Louis, MO, USA) for 60 min at 37 °C.

### 2.5. MTT (3-(4,5-Dimethylthiazol-2-yl)) and LDH (Lactate Dehydrogenase) Assay

PBMC CD14+ cells were cultured under osteoclast differentiation conditions. For MTT and LDH assay, EZ-Cytox (DoGen, Kyoto, Japan) and EZ LDH (DoGen, Kyoto, Japan) were used according to manufacturer’s instructions. On the indicated day, cells were treated, incubated for 60 min at 37 °C, and absorbance measured at 450 nm with a microplate reader.

### 2.6. RNA Extraction and Real-Time Quantitative (RT-q PCR)

RT-qPCR was carried out as previously described [16]. In brief, total RNA from cells was isolated using TRIzol (Invitrogen, Carlsbad, CA, USA), and transcribed to reverse transcription (Invitrogen, Carlsbad, CA, USA). qPCR was performed using CFX96 real-time PCE detection system (Bio-Rad Laboratories, Hercules, CA, USA). Expression of target genes was normalized to that of *GAPDH*. The primers used for qPCR are described in Table 1.

### 2.7. Immunoblotting (IB)

The immunoblotting procedure was previously described and followed [17,18]. The cells were lysed by 1× RIPA buffer containing 1× proteinase inhibitor cocktail (Calbiochem, Sam Diego, CA, USA), 1 mM PMSF (Sigma-Aldrich, St. Louis, MO, USA), and 1× phosphatase inhibitor (Cell Signaling, Danvers, MA, USA). The lysates were separated by SDS-PAGE on 10% gels, transferred to nitrocellulose membranes (GE Healthcare, Chicago, IL, USA), immunoblotted with primary and secondary antibodies, and visualized with ECL detection kits (Invitrogen, Carlsbad, CA, USA). Primary antibodies used for IB are described in Table 2.

### 2.8. Immunofluorescence

PBMC CD14+ cells were treated with Kyn under osteoclast differentiation conditions. At the end of the culture period, the cells were fixed with 10% formalin for 10 min and permeabilized with 1× PBS containing 0.3% Triton X-100 and 5% BSA for 1 h. The cells were incubated with primary antibody overnight and then Cy3- or Alexa 488-conjugated secondary antibody for 60 min. The stained cells were mounted with DAPI (Vector, Burlingame, CA, USA) and immunofluorescence images were obtained using a confocal microscope (Leica Microsystems, Wetzlar, Germany).

### 2.9. Transfection

PBMC CD14+ cells were treated with M-CSF (40 ng/mL) for 3 days. The cells were transfected with ON-TARGET plus SMARTpool siRNAs specific for AhR purchased from Dharmacon. ON-TARGET plus Non-targeting Pool was used as control. For transfection, lipofectamine 3000 reagent (Invitrogen, Carlsbad, CA, USA) was used according to the manufacturer’s instructions. After siRNA transfection, the cells were cultured in α-MEM with M-CSF (40 ng/mL) and RANKL (80 ng/mL) for 7 days to generate mature osteoclasts.

### 2.10. Statistical Analysis

Results are presented as mean ± standard deviation (SD). Two groups were determined by unpaired *t*-test. Three or more groups were analyzed by one-way or two-way analysis of variance (ANOVA) followed by Tukey’s or Sidak’s multiple comparison test (GraphPad Prism version 7, San Diego, CA, USA). *p*-values below 0.05 (* *p* < 0.05, ** *p* < 0.01, *** *p* < 0.001) were considered statistically significant.

## 3. Results

### 3.1. AhR Expression Changes in Human Osteoclast during Differentiation

We analyzed the relevance of AhR in human osteoclast differentiation. Differentiation into osteoclasts was checked by demonstrating increased expression of the osteoclast marker genes *Cathepsin K*, *TRAP*, and proteins NFATc1, c-Fos (Figure 1A,B). Expression of *AhR* mRNA gradually decreased during osteoclast differentiation (Figure 1A). In contrast, AhR protein expression initially increased, and then decreased (Figure 1B). To verify whether M-CSF or RANKL is important for osteoclast differentiation that could regulate AhR protein expression, human PBMC CD14+ cells were cultured with M-CSF in the presence of or absence of RANKL. We found that M-CSF and RANKL increased AhR protein expression in OCPs (Figure 1C). To confirm whether Kyn could activate AhR in human OCPs, we treated human OCPs with Kyn and measured the change in expression level of AhR-induced genes such as *CYP1B1*, *AHRR*, and *TIPARP* (Figure 1D). We found that Kyn treatment significantly induced the expression levels of *CYP1B1*, *AHRR*, and *TIPARP* in human OCPs, indicating that AhR is functionally active in human OCPs.

### 3.2. AhR Agonists Inhibit Osteoclast Differentiation in Human PBMC and RA SFMC CD14+ Cells

To explore whether AhR agonists regulate the osteoclastogenesis in human cells, PBMC CD14+ cells were treated with different concentrations of Kyn (0, 10, 100 μM) under osteoclast differentiation conditions. Furthermore, 100 μM Kyn strongly suppressed the formation of TRAP+ osteoclasts and F-actin ring that is essential of bone resorbing function in osteoclasts (Figure 2A). The mRNA expression of *TRAP* and *Cathephsin K*, markers of osteoclast differentiation, was significantly reduced (Figure 2B). We also confirmed that there are no cell viability and toxicity by kyn treatment (Appendix A). Consistent with Kyn treatment result, the number of TRAP+ osteoclasts and F-actin ring significantly decreased in the presence of other AhR agonists, 30 nM FICZ and 100 nM BaP (Figure 2C). We next examined the effects of AhR in osteoclast differentiation on the synovial fluid macrophages of RA patients. Consistent with the results above, Kyn, FICZ, and BaP inhibited osteoclast differentiation in RA SFMC CD14+ cells. (Figure 2D,E). In contrast, AhR agonists had no effect on mouse osteoclasts (Appendix A). These results suggest that AhR is a negative regulator of human osteoclast differentiation in physiological and pathological conditions.

### 3.3. Treatment of Kyn in the Early Stage of Osteoclastogenesis Effectively Suppresses Osteoclast Formation

To determine at which stage AhR agonists inhibit osteoclast differentiation, human PBMC CD4+ cells were treated with Kyn for the indicated time (Figure 3A). We found that the number of TRAP+ osteoclasts were markedly decreased by Kyn at an early stage (Figure 3B). As a result, we suggest that Kyn works effectively in the early stage of human osteoclast differentiation and that AhR is highly expressed (Figure 1).

### 3.4. Blockade of AhR Signaling Reverses Kyn-Induced Inhibition of Osteoclastogenesis

To explore whether Kyn inhibits the osteoclastogenesis through the AhR pathway in human cells, the cells were treated with Kyn in the presence or absence of an AhR antagonist or AhR siRNA during osteoclast differentiation. We used CH223191, known as an AhR antagonist, to block its nuclear translocation by binding AhR protein [19]. PBMC CD14+ cells (monocytes) or OCPs were pretreated with CH223191 for 3 h and then Kyn was added under osteoclast differentiation conditions. CH223191 diminished the inhibitory effects of Kyn on osteoclastogenesis (Figure 4A). Additionally, the mRNA expression of *Cathepsin K* and *TRAP* decreased by Kyn was partially recovered by CH223191 (Figure 4B). To confirm that Kyn inhibits osteoclast differentiation via AhR signaling, human OCPs were transfected with AhR siRNA. The genetic knock down of AhR abolished AhR protein and the AhR-induced genes such as *CYP1B1*, *AHRR*, and *TIPARP* (Figure 4C,D). Subsequently, we induced the differentiation of siRNA-transfected cells into osteoclasts using M-CSF and RANKL in the presence or absence of Kyn. Consistent with the CH223191 treatment result, AhR siRNA diminished the Kyn-induced inhibition of osteoclastogenesis (Figure 4E). AhR siRNA also reversed Kyn-induced inhibition of *Cathepsin K* and *TRAP* expression (Figure 4F). Taken together, these results indicate that Kyn inhibits the osteoclastogenesis through AhR signaling.

### 3.5. Kyn Does Not Regulate RANKL-Induced Signaling in OCPs

The osteoclast differentiation is dependent on the MAPK and NF-κB pathways activated by RANKL, and we therefore investigated whether Kyn inhibits RANKL-induced signaling. We pretreated human OCPs with Kyn for 3 h and exposed them to RANKL for the indicated time. MAPKs and NF-κB pathways were activated by RANKL in human OCPs, and Kyn had no effect on RANKL-induced activation of these pathways (Figure 5). Thus, these data indicate that Kyn-induced suppression of osteoclastogenesis is not mediated by alteration of RANKL signaling.

### 3.6. Kyn Downregulates NFATc1 Protein in Human Osteoclasts

To explore how Kyn inhibits RANKL-induced osteoclast differentiation in human cells, we first measured the changes in the mRNA expression of *Myc*. Myc is required for osteoclastogenesis [20], and AhR was reported to suppress c-Myc expression [21]. Thus, we tested whether Kyn suppresses *Myc* mRNA expression in human osteoclasts. Our result showed that Kyn does not affect *Myc* expression in human osteoclasts (Figure 6A). Next, we measured the changes of *ARNT* and *NFATc1* mRNA expression. ARNT acts as a transcriptional activator by binding with AhR, and indoxyl-sulfate (IS, an AhR agonist) inhibits osteoclast differentiation by regulating NFATc1 and ARNT expression [22]. However, Kyn did not affect *ARNT* and *NFATc1* mRNA expression in human cells (Figure 6A).

Next, we measured the Syk and RelA/p65 protein expression. Activation of AhR was reported to induce the degradation of Syk and RelA/p65 protein [23,24]. However, Kyn did not inhibit the expression of Syk and RelA/p65 protein in human cells (Figure 6B).

NFATc1 is a critical transcription factor for osteoclast differentiation. Interestingly, Kyn treatment inhibited the NFATc1 protein level without changes in the mRNA level during osteoclast differentiation (Figure 6A,B). We also observed nuclear localization of NFATc1 protein at mature osteoclasts, and Kyn treatment inhibited this nuclear expression (Figure 6C). Moreover, the inhibition of Kyn on the protein expression of NFATc1 was partially reversed by CH223191 or AhR siRNA (Figure 6E,G). However, we found that *NFATc1* mRNA expression was not affected by Kyn (Figure 6D,F), CH223191 or AhR siRNA, suggesting that Kyn regulates NFATc1 protein expression, without a change in mRNA levels in human osteoclasts (Figure 6D–G). Activation of AhR was reported to suppress the protein expression by inducing ubiquitin-proteosomal and lysosomal degradation. Therefore, we examined whether Kyn degrades NFATc1 protein in human osteoclasts using MG132, an inhibitor of the proteasome, or CQ, a lysosome inhibitor. The results showed that neither MG132 nor CQ affect the inhibition of Kyn on NFATc1 protein expression (Figure 6H,I).

## 4. Discussion

In summary, we found that AhR protein expression initially increased in the early stage of human osteoclast differentiation and decreased in mature osteoclasts. AhR agonists, such as Kyn, FICZ, and BaP, led to osteoclastogenesis inhibition in human cells and this inhibitory effect was observed in the early stage of osteoclast differentiation where AhR is highly expressed. AhR siRNA or CH223191 diminished the Kyn-induced inhibition of osteoclast differentiation in human cells, indicating that AhR is involved in the anti-osteoclastogenic effect of Kyn. Kyn inhibits the protein expression of NFATc1, a key transcription factor in osteoclast differentiation, but not *NFATc1* mRNA in human osteoclast, suggesting that Kyn inhibits human osteoclastogenesis through downregulation of NFATc1 protein expression.

AhR is a ligand-activated transcription factor and plays an important role in bone remodeling. Recent studies show a relatively consistent explanation for the effects of AhR on osteoblasts. The overall effect of AhR activation in osteoblasts is inhibited cell differentiation. However, the effects of AhR on osteoclast differentiation and function remain unclear [12,13]. AhR induces intracellular signaling via either the classical genomic or the non-genomic pathways. The translocation of the AhR complex to the nucleus is induced by interaction with AhR agonist and AhR controls gene transcription by binding DRE containing genes. In addition, AhR activates non-genomic pathways, such as the release of c-src kinase, with consequent phosphorylation of target proteins and functioning as an E3 ubiquitin ligase. Depending on whether AhR functions as a ligand-activated transcription factor or an E3 ubiquitin ligase, it either promotes or inhibits osteoclast differentiation. AhR promotes osteoclast differentiation and function through activation of RANK/c-Fos signaling and NF-κB signaling, or induction of Blimp1, Cyp1b1, and Cyp1a2 expression [25,26,27]. In contrast, AhR inhibits osteoclast differentiation in the non-genomic pathway through proteasomal degradation of NFATc1 and Syk [23]. More recently, it was reported that the AhR genomic pathway is involved in a Kyn-induced increase of RANKL-induced osteoclastogenesis [28]. However, all studies published so far on the regulation of osteoclast differentiation by AhR were conducted in mouse cells, not in human cells. In this study, we show that AhR is an inhibitory molecule in human osteoclast differentiation.

Notably, marked differences were reported in the regulation of osteoclast differentiation and function in mice and humans. For instance, IL-27 acts as a potent inhibitory molecule of human OCPs, whereas in mice, IL-27 has a minimal-to-moderate effect on osteoclast differentiation [29]. Furthermore, in humans, triggering receptors expressed on myeloid cells 2 (TREM-2) deficiency inhibited osteoclast differentiation and bone remodeling, whereas in mice, TREM-2 deficiency has unclear effects in vivo and has increased osteoclastogenesis in vitro [30]. In a previous report, *AhR* mRNA expression increased after RANKL treatment in mouse bone marrow macrophages (BMMs). Consistent with increased *AhR* mRNA expression, RANKL induced an increase of AhR expression in the cytoplasm and nucleus of mouse BMMs [26]. In contrast to these previous results, our study showed that expression of the mRNA level of *AhR* gradually decreased during human osteoclast differentiation. Furthermore, AhR protein expression initially increased, and then decreased during human osteoclast maturation.

To explore how Kyn inhibits RANKL-induced osteoclast differentiation in human cells, we measured the Syk, RelA/p65, and NFATc1 protein expressions. AhR inhibits mouse osteoclast differentiation and function through several different mechanisms. One well-known and important mechanism is that AhR functions as an E3 ubiquitin ligase that induces the proteasomal degradation of osteoclastogenesis-essential proteins, such as Syk and NFATc1 [23]. In addition, activation of AhR induces ubiquitin-proteosomal and lysosomal degradation of RelA/p65, a key classical NF-κB pathway protein [24]. We showed that Kyn did not inhibit the expression of Syk and RelA/p65 protein in human OPCs. Additionally, Kyn did not alter RANKL-induced MAPK and NF-κB signaling in human cells. However, Kyn inhibits the protein expression of NFATc1, but not *NFATc1* mRNA in human cells. In a previous study using Raw 264.7 cells, indoxyl sulfate (IS), an AhR agonist, induces NFATc1 ubiquitination and proteasomal degradation [22]. In this study, the gene expression of *NFATc1* increased at low doses of IS and decreased at high doses of IS. In addition, they showed that AhR siRNA suppresses the gene expression of *NFATc1*, suggesting that *NFATc1* gene expression is regulated by AhR. In contrast, our study shows that the AhR agonist does not affect *NFATc1* gene expression in human cells.

AhR agonists generated from natural products have emerged as potential targets for RA treatment [31]. For example, leflunomide is clinically used for RA treatment, indicating that AhR activation attenuates bone erosion of RA [32]. In contrast, an increased AhR expression was detected in synovial tissue of patients with RA compared to osteoarthritis patients [33], indicating that AhR is associated with the pathophysiology of RA. Although association of AhR with RA is still a controversial issue, regulating AhR signaling in RA could be a therapeutic target for bone destruction by osteoclasts.

Our study has several limitations. First, we suggest that AhR activation inhibits osteoclast differentiation through the suppression of NFATc1 protein expression in human cells, the mechanism of AhR-induced suppression of NFATc1 protein expression remains unknown. Second, marked differences were observed in the regulation of osteoclast differentiation and function in mice and humans in the context of AhR modulation, it is impossible to confirm the in vivo effects of AhR on humans using specific disease model mice.

## 5. Conclusions

AhR activation inhibits osteoclast differentiation in human cells, and the suppressed NFATc1 protein expression may be involved in this inhibitory effect of AhR. Our study highlights AhR as a strong potential therapeutic molecule in human diseases in which osteoclast activation play important pathogenic roles.

## Figures and Tables

**Figure 1 cells-10-03498-f001:**
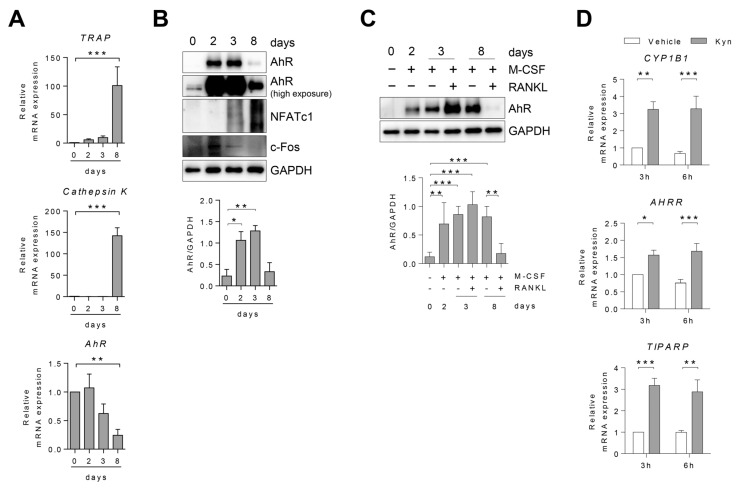
The expression of AhR changes during osteoclastogenesis. Human PBMC CD14+ cells were cultured with M-CSF (20 ng/mL) for 2 days, and then RANKL (40 ng/mL) was added for 6 days. (**A**) *AhR*, *Cathepsin K*, and *TRAP* mRNA expression was measured using RT-qPCR and normalized to that of *GAPDH*. (**B**,**C**) Level of AhR protein was detected by immunoblotting and measured by Image J. (**D**) OCPs were treated with Kyn (100 μM) for 3 h or 6 h. AhR-induced genes of *CYP1B1*, *AHRR*, and *TIPARP* were analyzed using RT-qPCR and normalized to that of GAPDH. Data are shown as mean ± SD of more than five independent experiments and one-way or two-way ANOVA tests were applied. * *p* < 0.05, ** *p* < 0.01, *** *p* < 0.001.

**Figure 2 cells-10-03498-f002:**
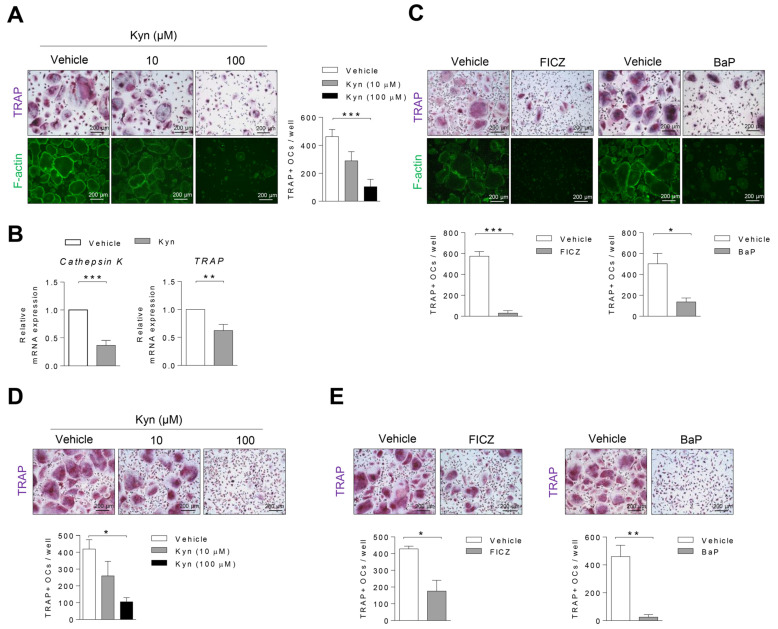
AhR agonists treatment inhibits osteoclast differentiation. Human PBMC CD14+ cells were cultured with (**A**) Kyn (0, 10, 100 μM), (**C**) FICZ (0, 30 nM), or BaP (0, 100 nM) for 8 days under osteoclast differentiation conditions. The cells were stained with TRAP and F-actin ring on day 8, and the TRAP+ osteoclasts were counted. (**B**) Mature osteoclast markers of *Cathepsin K* and *TRAP* mRNA expression were measured using RT-qPCR and normalized to that of *GAPDH*. (**D**,**E**) Human RA SFMC CD14+ cells were cultured with (**D**) Kyn, (**E**) FICZ, or BaP for 9 days under osteoclast differentiation conditions. On day 9, the cells were stained with TRAP, and the TRAP+ osteoclasts were counted. Images were taken at original magnification 100 X. Scale bar is 200 μm. Data are shown as mean ± SD of more than three independent experiments and a one-way ANOVA test or unpaired *t*-test were applied. * *p* < 0.05, ** *p* < 0.01, *** *p* < 0.001.

**Figure 3 cells-10-03498-f003:**
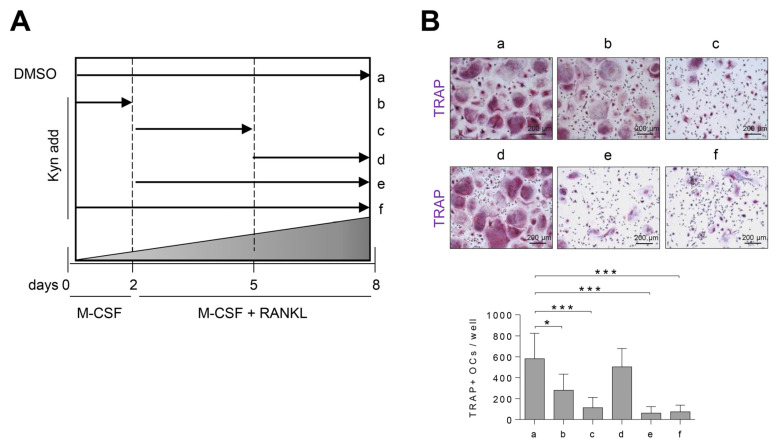
Kyn suppresses osteoclast formation at an early stage of differentiation. (**A**,**B**) In the absence (a) or presence (b–f) of Kyn (100 μM) for the indicated time, human PBMC CD14+ cells were cultured with M-CSF (20 ng/mL) for 2 days, and then RANKL (40 ng/mL) was added for 6 days. The cells were stained with TRAP on day 8, and the TRAP+ osteoclasts were counted. Images were taken at original magnification 100 X. Scale bar is 200 μm. Data are shown as mean ± SD of five independent experiments and a one-way ANOVA test was applied. * *p* < 0.05, *** *p* < 0.001.

**Figure 4 cells-10-03498-f004:**
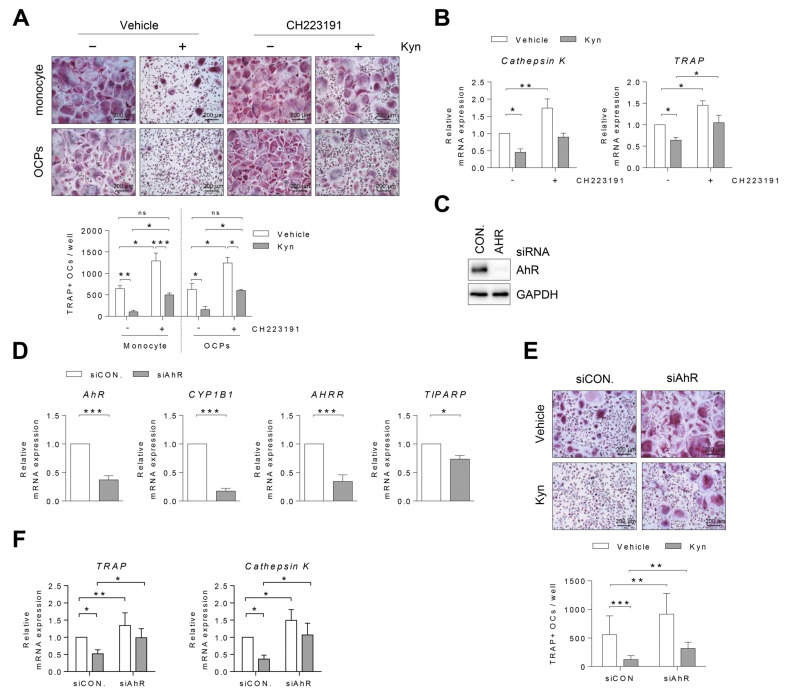
Blockade of AhR signaling reverses Kyn-induced inhibition of osteoclast differentiation. (**A**) Human PBMC CD14+ cells (monocytes) or OCPs were pretreated with CH223191 (5 μM) for 3 h and then cultured with Kyn (100 μM) under osteoclast differentiation conditions. On day 8, the cells were stained with TRAP and the TRAP+ osteoclasts were counted. (**B**) Mature osteoclast markers of *Cathepsin K* and *TRAP* mRNA expression were measured using RT-qPCR and then normalized to that of *GAPDH*. (**C**,**D**) Human PBMC CD14+ cells were treated with M-CSF (40 ng/mL) for 3 days, and then transfected with siRNA against control (siCON.) or AhR (siAhR) for 24 h. (**C**) Level of AhR protein was detected by immunoblotting. (**D**) AhR-induced genes of *CYP1B1*, *AHRR*, and *TIPARP* were analyzed using RT-qPCR and normalized to that of *GAPDH*. (**E**,**F**) After transfection, the cells were cultured with Kyn (100 μM) in the presence of M-CSF (40 ng/mL) and RANKL (80 ng/mL) for 7 days. (**E**) The cells were stained with TRAP on day 7, and the TRAP+ osteoclasts were counted. (**F**) Mature osteoclast markers of *Cathepsin K* and *TRAP* mRNA expression were measured using RT-qPCR and normalized to that of *GAPDH*. Images were taken at original magnification 100×. Scale bar is 200 μm. Data are shown as mean ± SD of more than three independent experiments and a two-way ANOVA test was applied. * *p* < 0.05, ** *p* < 0.01, *** *p* < 0.001, ns *p* > 0.05.

**Figure 5 cells-10-03498-f005:**
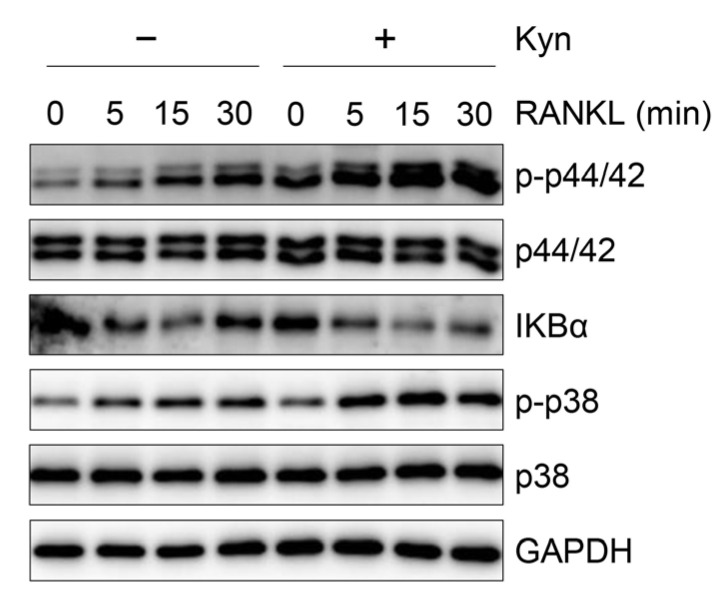
Kyn does not inhibit RANKL-induced signaling in OCPs. Human PBMC CD14+ cells were cultured with M-CSF (20 ng/mL) for 2 days. The OCPs were treated with Kyn (100 μM) for 3 h, and then the cells were stimulated with RANKL (40 ng/mL) for the indicated time.

**Figure 6 cells-10-03498-f006:**
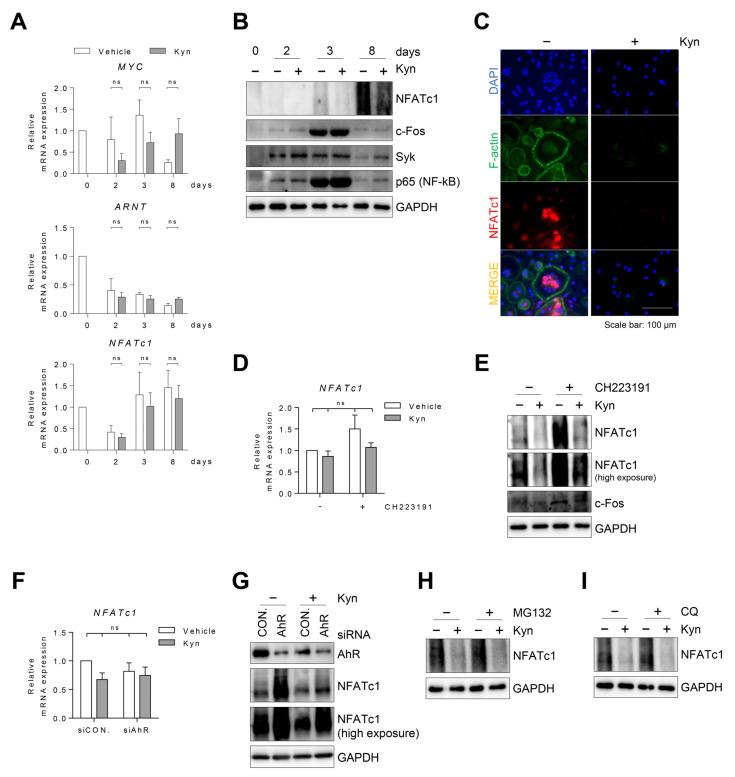
Kyn reduces NFATc1 protein in osteoclasts. Human PBMC CD14+ cells were cultured with Kyn (100 μM) under osteoclast differentiation conditions. (**A**) The mRNA expression of *Myc*, *ARNT*, and *NFATc1* was measured using RT-qPCR and then normalized to that of *GAPDH*. (**B**) Level of NFATc1, c-Fos, Syk, and p65 proteins was detected by immunoblotting. (**C**) Immunostaining with NFATc1 (red), F-actin (green), and DAPI (blue) were observed. Scale bar is 100 μm. (**D**,**E**) Human PBMC CD14+ cells were pretreated with CH223191 (5 μM) for 3 h and then cultured with Kyn (100 μM) under osteoclast differentiation conditions. On day 8, (**D**) mRNA expression and (**E**) protein level of NFATc1 were analyzed by RT-qPCR and immunoblotting. (**F**,**G**) After transfection, (**F**) mRNA expression and (**G**) protein level of NFATc1 were analyzed by RT-qPCR and immunoblotting on day 7. (**H**,**I**) Human PBMC CD14+ cells were cultured with Kyn (100 μM) under osteoclast differentiation conditions. On day 8, MG132 (10 μM) or CQ (50 μM) was added in the osteoclasts for 6 h. Level of NFATc1 protein was detected by immunoblotting. Data are shown as mean ± SD of more than three independent experiments and a two-way ANOVA test was applied. ns *p* > 0.05.

**Table 1 cells-10-03498-t001:** RT-qPCR primer sequences.

Gene	Forward Primer (5′→3′)	Reverse Primer (5′→3′)
*AhR*	ACATCACCTACGCCAGTCG	TCTATGCCGCTTGGAAGGAT
*Cathepsin K*	CTCTTCCATTTCTTCCACGAT	ACACCAACTCCCTTCCAAAG
*TRAP*	TGGCTTTGCCTATGTGGA	CCTGGTCTTAAAGAGGGACTT
*NFATc1*	GCATCACAGGGA AGACCGTGTC	GAAGTTCAATGTCGGAGTTTCTGAG
*CYP1B1*	AGTTCTTGAGGCACTGCGAA	GTGATAGTGGCCGGTACGTT
*AHRR*	ACCGCGGATGCAAAAGTAAAAG	CTCCTTCCTGCTGAGTAATTG
*TIPARP*	CACCCTCTAGCAATGTCAACTC	AGACTCGGGATACTCTCTCC
*C-Myc*	CAGCGAGGATATCTGGAAGA	CTTCTCTGAGACGAGCTT
*ARNT1*	ACTACCCGCTCAGGCTTTTC	ATGGAGTCTGAAAGCTGCCC
*GAPDH*	CAAGATCATCAGCAATGCC	CTGTGGTCATGAGTCCTTCC

**Table 2 cells-10-03498-t002:** Primary antibodies used in IB.

Antigen	Manufacturer	Catalog Number
AhR	Cell Signaling, Danvers, MA, USA	83200
NFATc1	BD biosciences, San Jose, CA, USA	556602
c-Fos	Abcam, Cambridge, MA, USA	ab190289
p-p44/42	Cell Signaling, Danvers, MA, USA	9101
p44/42	Cell Signaling, Danvers, MA, USA	9102
IKBα	Cell Signaling, Danvers, MA, USA	9242
p-p38	Cell Signaling, Danvers, MA, USA	9215
p38	Santa Cruz, Dallas, CA, USA	535
Syk	Cell Signaling, Danvers, MA, USA	2712
p65	Santa Cruz, Dallas, CA, USA	372
GAPDH	Cell Signaling, Danvers, MA, USA	2118

## Data Availability

Not applicable.

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
