# Peer review of "Inhibition of Human Osteoclast Differentiation by Kynurenine through the Aryl-Hydrocarbon Receptor Pathway"

_cells, 2021, doi:10.3390/cells10123498_

Round 1
Reviewer 1 Report
In this work So Yeon Kim focused the study on the regulation of AhR-hydrocarbon receptor pathway in human osteoclast differentiation, given its involvement on bone remodeling. In particular, the author has shown that AhR is highly expressed in the early stage of osteoclastogenesis and decreased in mature osteoclasts. Interestingly, Kynurenine (Kyn), which is an AhR agonist, inhibited osteoclast formation and suppressed osteoclast differentiation. The experimental design is well written, and the type of experiments have been properly chosen to support the main claims of the paper. However, in some cases statistical analysis are missing and only descriptive sentences are given.
More principally, I have the following minor concerns with the manuscript:
Figure 1 B: Even though the changes in AhR protein expression during osteoclastogenesis are obvious, statistical analysis is required.
Figure 1 C: It is shown that AhR protein level increases between day 2 (second lane, only M-CSF) and day 3 (4th lane, M-CSF+RANKL). Looking the western blot, this increase is quite strong. However, in Figure 1B between day 2 and 3 there are no difference (again, without statistical analysis this is the only conclusion) despite the protocol used to push the differentiation is the same. Could the authors explain these apparent discrepancies?
Figure 2 A: During osteoclast differentiation, PBMC CD14+ cells were treated with different concentrations of Kyn (0, 10, 100 μM). It is shown that 100 μM strongly suppressed the formation of TRAP+ osteoclasts. Did the authors test whether this concentration has any cytotoxic effect in their experimental condition?
Figure 4B: TRAP mRNA expression tended to decrease (not statistically significant) by Kyn and rescued by CH22391, which was not significant. However, in Figure 2C the Kyn treatment significantly reduced TRAP level. At this point it is not clear if the result with CH22391 (failed rescue) was hampered by the fact that the Kyn treatment didn’t work in this experimental setting. In other words, the control treatment (Kyn) didn’t work. The authors should repeat the experiments or clarify these discrepancies.
Minor concerns:
Figure 4. All the starred bars in Figure 4 have the vertical lines misaligned (same problem in other figures). Please check and correct it.
Author Response
We are very grateful for the reviewer’s positive comments and suggestions to improve the impact of our study. In response to the points and suggestion raised by the reviewer, we have substantially revised our paper. Point-by-point response to the reviewer’s comments is marked in RED as follow:
Reviewer 1
In this work So Yeon Kim focused the study on the regulation of AhR-hydrocarbon receptor pathway in human osteoclast differentiation, given its involvement on bone remodeling. In particular, the author has shown that AhR is highly expressed in the early stage of osteoclastogenesis and decreased in mature osteoclasts. Interestingly, Kynurenine (Kyn), which is an AhR agonist, inhibited osteoclast formation and suppressed osteoclast differentiation. The experimental design is well written, and the type of experiments have been properly chosen to support the main claims of the paper. However, in some cases statistical analysis are missing and only descriptive sentences are given.
More principally, I have the following minor concerns with the manuscript:
Figure 1 B: Even though the changes in AhR protein expression during osteoclastogenesis are obvious, statistical analysis is required.
RESPONSE: As requested, we added to figure 1B. Please see the revised figure 1B.
Figure 1 C: It is shown that AhR protein level increases between day 2 (second lane, only M-CSF) and day 3 (4th lane, M-CSF+RANKL). Looking the western blot, this increase is quite strong. However, in Figure 1B between day 2 and 3 there are no difference (again, without statistical analysis this is the only conclusion) despite the protocol used to push the differentiation is the same. Could the authors explain these apparent discrepancies?
RESPONSE: We truly appreciated your constructive comment. Based on the comment, we quantified the AhR protein quantification data and added it in the REVISED figure 1B. Immunoblotting data for figure 1C seems like the increase, but the quantification graph of whole data was not statistically significant between them. Since there are no difference, we did not mark it in the REVISED figure 1C.
Figure 2 A: During osteoclast differentiation, PBMC CD14+ cells were treated with different concentrations of Kyn (0, 10, 100 μM). It is shown that 100 μM strongly suppressed the formation of TRAP+ osteoclasts. Did the authors test whether this concentration has any cytotoxic effect in their experimental condition?
RESPONSE: Thanks for the comment. Based on the request, we performed MTT and LDH assay in presence and absence of Kyn during osteoclastogenesis and added the data to REVISED figure S1. Please see the 3.2 result section of 6 page in the revised manuscript.
Figure 4B: TRAP mRNA expression tended to decrease (not statistically significant) by Kyn and rescued by CH22391, which was not significant. However, in Figure 2C the Kyn treatment significantly reduced TRAP level. At this point it is not clear if the result with CH22391 (failed rescue) was hampered by the fact that the Kyn treatment didn’t work in this experimental setting. In other words, the control treatment (Kyn) didn’t work. The authors should repeat the experiments or clarify these discrepancies.
RESPONSE: We truly appreciate this appropriate point. In the REVISED figure 2B (figure 2C of previous version), 6 experiments were performed, but in figure 4B, only 3 were performed. Due to the large genetic differences between donors, meaningful statistics could not be obtained with just 3 experiments. We added two experiments and obtained significant results.
Minor concerns:
Figure 4. All the starred bars in Figure 4 have the vertical lines misaligned (same problem in other figures). Please check and correct it.
RESPONSE: We modified all the starred bars.
Reviewer 2 Report
The authors investigated the role of Aryl-hydrocarbon receptor (AhR) on osteoclastogenesis. They tested the effect of AhR agonists and antagonists using human osteoclast precursors. They found that AhR negatively regulates human osteoclast differentiation via modulating NFATc1 protein expression.
The study is fairly conducted, and the manuscript is well written.
I have some comments to improve the manuscript.
Major concerns
#1. They proposed that AhR signaling decreases osteoclastogenesis via reduced NFATc1 protein expression without affecting NFATc1 mRNA expression. Since autoamplification and nuclear translocation of NFATc1 are essential on osteoclastogenesis, I would suggest them clarify these mechanisms in their experiments. In terms of autoamplification, Asagiri M et al.( J Exp Med. 2005 Nov 7;202(9):1261-9) reported that a short isoform of NFATc1 is amplified during osteoclastogenesis. The authors should test the gene expression of the short isoform. Also, the protein expression of the short isoform can be distinguished by immunoblotting, depending on the detection antibody. The authors should carefully interpret the bands in their experiments. In terms of nuclear localization, immunoblotting using nuclear fraction protein or immunocytostaining should be performed.
#2. Although they propose “AhR could be good therapeutic target to prevent bone destruction in chronic inflammatory diseases such as rheumatoid arthritis (RA)” in the abstract, this point is not substantially discussed in the manuscript. They should add one paragraph in the discussion section and discuss the possibility of targeting AhR signaling in pathological conditions. They should describe whether the expression of AhR itself or activation of related pathways are affected in some pathological conditions, such as RA and osteoporosis. Also, possible strategies should be described, e.g., candidates to increased endogenous AhR agonists, if any.
#3. The mode of action of CH223191 should be described to interpret the experimental data with Kyn and CH223191.
#4. The tested concentrations of Kyn might have some toxic effect on the cells, resulting in decreased osteoclastogenesis. To exclude the possibility, the toxicity of Kyn should be evaluated, e.g., proliferation/viability assay or measuring cell death or LDH.
#5. In a comparison of their data, they should compare “Kyn(-), CH223191(-)“ vs. “Kyn(-), CH223191(+)” in Figure 4B. Similar comparisons should be done and add statistical results in each graph in other similar experiments.
Author Response
We are very grateful for the reviewer’s positive comments and suggestions to improve the impact of our study. In response to the points and suggestion raised by the reviewer, we have substantially revised our paper. Point-by-point response to the reviewer’s comments is marked in RED as follow:
Reviewer 2
The authors investigated the role of Aryl-hydrocarbon receptor (AhR) on osteoclastogenesis. They tested the effect of AhR agonists and antagonists using human osteoclast precursors. They found that AhR negatively regulates human osteoclast differentiation via modulating NFATc1 protein expression.
The study is fairly conducted, and the manuscript is well written.
I have some comments to improve the manuscript.
Major concerns
#1. They proposed that AhR signaling decreases osteoclastogenesis via reduced NFATc1 protein expression without affecting NFATc1 mRNA expression. Since autoamplification and nuclear translocation of NFATc1 are essential on osteoclastogenesis, I would suggest them clarify these mechanisms in their experiments. In terms of autoamplification, Asagiri M et al.( J Exp Med. 2005 Nov 7;202(9):1261-9) reported that a short isoform of NFATc1 is amplified during osteoclastogenesis. The authors should test the gene expression of the short isoform. Also, the protein expression of the short isoform can be distinguished by immunoblotting, depending on the detection antibody. The authors should carefully interpret the bands in their experiments. In terms of nuclear localization, immunoblotting using nuclear fraction protein or immunocytostaining should be performed.
RESPONSE: We appreciate valuable comments and suggestions. The above-mentioned study (Asagiri M et al. J Exp Med. 2005 Nov 7;202(9):1261-9) was conducted in mouse. However, our study conducted in human cells. Chrisopher J. Day et al. (J Cell Biochem. 2005 May 1;95(1):17-23) reported that the high molecular weight isoforms of NFATc1 was increased during human osteoclastogenesis. To compare NFATc1 protein expression in human cells, we performed overexpression with human NFATc1 gene in HEK293T cell. We observed that the transfected NFATc1 protein is expressed in high molecular weight isoforms (around 100kDa at protein size markers), and the NFATc1 protein was also decreased by Kyn. You can see the below. As requested by reviewer, we newly added the data on nuclear localization of NFATc1 protein at mature osteoclasts. Please see the REVISED figure 6C.
#2. Although they propose “AhR could be good therapeutic target to prevent bone destruction in chronic inflammatory diseases such as rheumatoid arthritis (RA)” in the abstract, this point is not substantially discussed in the manuscript. They should add one paragraph in the discussion section and discuss the possibility of targeting AhR signaling in pathological conditions. They 1) should describe whether the expression of AhR itself or activation of related pathways are affected in some pathological conditions, such as RA and osteoporosis. Also, 2) possible strategies should be described, e.g., candidates to increased endogenous AhR agonists, if any.
RESPONSE: We described in the discussion section. Please see the third paragraph of 12 page in the revised manuscript.
#3. The mode of action of CH223191 should be described to interpret the experimental data with Kyn and CH223191.
RESPONSE: We described it and marked as red in the 3.4 result section. Pleases see the 8 page in the revised manuscript.
#4. The tested concentrations of Kyn might have some toxic effect on the cells, resulting in decreased osteoclastogenesis. To exclude the possibility, the toxicity of Kyn should be evaluated, e.g., proliferation/viability assay or measuring cell death or LDH.
RESPONSE: Thanks for the comment. Based on the request, we performed MTT and LDH assay in presence and absence of Kyn during osteoclastogenesis and added the data to REVISED figure S1. Please see the 3.2 result section of 6 page in the revised manuscript.
#5. In a comparison of their data, they should compare “Kyn(-), CH223191(-)“ vs. “Kyn(-), CH223191(+)” in Figure 4B. Similar comparisons should be done and add statistical results in each graph in other similar experiments.
RESPONSE: Thanks for the comment. As requested, we carefully revised the statistical results of all experimental data and applied to the REVISED figures and manuscript.

Reviewer 3 Report
The study is devoted to the evaluation of Aryl-hydrocarbon receptor (AhR) function in osteoblast differentiation in in vitro human model. The results show that, in contrast to the murine data previously obtained, in human model AhR agonists treatment inhibits osteoclast differentiation. AhR agonist Kynurenine (Kyn) suppresses osteoclast formation at an early stage of differentiation and blockade of AhR signaling reverses Kyn-induced inhibition of osteoclast differentiation. Finally, it is shown that AhR is a negative regulator in differentiation of human osteoclast.
The authors have to address the following minor comments:
(1) The origin country of the all used materials and facilities in sec.2 should be given.
(2) All abbreviations (other than the generally recognized ones) must be deciphered at their first mention in the text.
(3) Bars on microscopic images should be made clearly visible
Author Response
We are very grateful for the reviewer’s positive comments and suggestions to improve the impact of our study. In response to the points and suggestion raised by the reviewer, we have substantially revised our paper. Point-by-point response to the reviewer’s comments is marked in RED as follow:
Reviewer 3
The study is devoted to the evaluation of Aryl-hydrocarbon receptor (AhR) function in osteoblast differentiation in in vitro human model. The results show that, in contrast to the murine data previously obtained, in human model AhR agonists treatment inhibits osteoclast differentiation. AhR agonist Kynurenine (Kyn) suppresses osteoclast formation at an early stage of differentiation and blockade of AhR signaling reverses Kyn-induced inhibition of osteoclast differentiation. Finally, it is shown that AhR is a negative regulator in differentiation of human osteoclast.
The authors have to address the following minor comments:
- The origin country of the all used materials and facilities in sec.2 should be given.
RESPONSE: Thanks for the comment. As requested, we edited it all throughout the manuscript.
- All abbreviations (other than the generally recognized ones) must be deciphered at their first mention in the text.
RESPONSE: Thanks for the comment. As requested, we edited it all throughout the manuscript.
- Bars on microscopic images should be made clearly visible
RESPONSE: Thanks for the comment. As requested, we modified the scale bar to be more visible in the images.
Round 2
Reviewer 2 Report
The manuscript is appropriately revised following the suggestions.
I have no additional comment.